# Structural basis for ELL2 and AFF4 activation of HIV-1 proviral transcription

Shiqian Qi[1,2], Zichong Li[2], Ursula Schulze-Gahmen[2], Goran Stjepanovic[2,3], Qiang Zhou[2] & James H. Hurley[2,3]

The intrinsically disordered scaffold proteins AFF1/4 and the transcription elongation factors ELL1/2 are core components of the super elongation complex required for HIV-1 proviral transcription. Here we report the 2.0-Å resolution crystal structure of the human ELL2 C-terminal domain bound to its 50-residue binding site on AFF4, the ELLBow. The ELL2 domain has the same arch-shaped fold as the tight junction protein occludin. The ELLBow consists of an N-terminal helix followed by an extended hairpin that we refer to as the elbow joint, and occupies most of the concave surface of ELL2. This surface is important for the ability of ELL2 to promote HIV-1 Tat-mediated proviral transcription. The AFF4–ELL2 interface is imperfectly packed, leaving a cavity suggestive of a potential binding site for transcription-promoting small molecules.

[1] Department of Urology, State Key Laboratory of Biotherapy, West China Hospital, Sichuan University and National Collaborative Innovation Center, Chengdu 610041, China. [2] Department of Molecular and Cell Biology and California Institute of Quantitative Biosciences, University of California, Berkeley, Berkeley, California 94720, USA. [3] Molecular Biophysics and Integrated Bioimaging Division, Lawrence Berkeley National Laboratory, Berkeley, California 94720, USA. Correspondence and requests for materials should be addressed to J.H.H. (email: jimhurley@berkeley.edu).

Curing AIDS is a major global health goal. AIDS is caused by the human immunodeficiency virus (HIV), which has proved exceptionally difficult to eradicate[1,2]. The principal obstacle to HIV eradication is the persistence in patients of a reservoir of cells harbouring latent provirus integrated within the genome[3]. Clinical interest in the reactivation of latent HIV[1,2] has brought renewed attention to the mechanism of transcriptional regulation of the HIV provirus. Latency is regulated at the levels of epigenetic silencing, and transcription initiation and elongation[4]. Transcription elongation, which is promoted by the HIV Tat protein and TAR RNA sequence, is the best understood of these mechanisms. The functions of HIV Tat and TAR in promoting elongation are completely dependent on their ability to hijack the host super elongation complex (SEC)[5–7].

The SEC consists of the Cyclin-dependent kinase CDK9 and Cyclin T (CycT1 or T2), together known as P-TEFb[8]; one of either of the intrinsically disordered (ID) scaffold proteins AFF1 or AFF4 (refs 5,6); one of either ENL or AF9; and one of either of the RNA polymerase elongation factors ELL1 or ELL2 (refs 5,9,10). The reason that Tat is such a powerful activator of HIV-1 transcription lies in its ability to pack two distinct transcriptional elongation factors P-TEFb and ELL1/2 into a single SEC complex, where the two factors can synergistically stimulate a single RNA Pol II elongation complex[5,7]. AFF1/4 is >1,100 residues long and is the principal scaffold that holds the SEC together[11]. AFF1/4 consists almost entirely of predicted ID regions (IDRs). AFF1 and AFF4 function in transcription elongation by virtue of various peptide motifs interspersed throughout their sequences, much like many other ID signalling and regulatory proteins that have come under intensive study[12,13]. The AFF1- and ELL2-containing version of the SEC is the most important in the promotion of proviral elongation, despite its low abundance[14].

The structure of P-TEFb lacking the C-terminal IDR of CycT1 has been determined in complex with HIV-1 Tat[15] and the N-terminal 60 residues of AFF4 (refs 16,17). This structure shows that AFF4 residues 32–67 bind as an extended strand

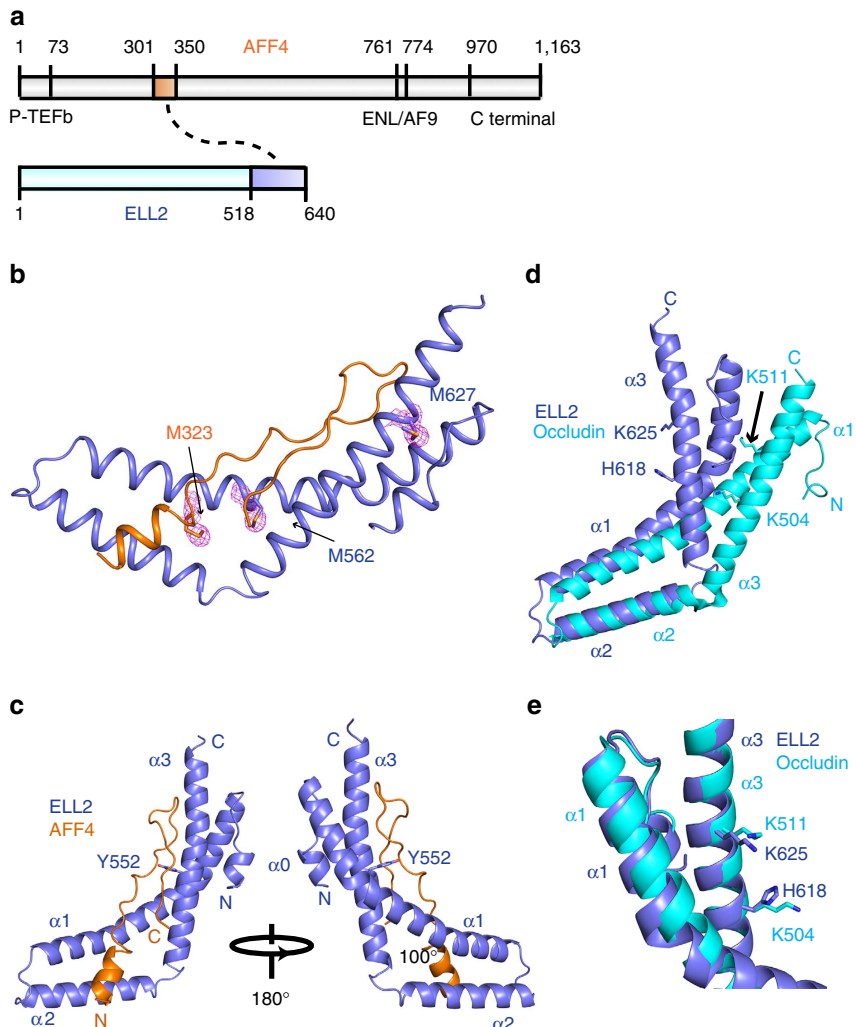

**Figure 1 | Crystal structure of the AFF4 ELLBow in complex with the occludin homology domain of ELL2. (a)** Schematic of the interactions of the AFF4 intrinsically disordered protein (IDP) scaffold with its partners in the SEC. The highlighted boxes within AFF4 and ELL2 represent the co-crystallized elements described below. Other regions of AFF4 are annotated for binding to P-TEFb, AF9/ENL and the novel C-terminal ELL1/2-binding site described below. **(b)** Se anomalous difference peaks and overall structure of the complex. The Se substructure map is displayed at a contour level of 2σ(magenta). **(c)** Two views of the overall structure of the complex, with AFF4 in orange and ELL2 in light blue. **(d,e)** Comparison of ELL2 and occludin C-terminal domain showing that the folds are similar, but ELL2 is more sharply bent. Helix α3 from both ELL2 and occludin are aligned, the structurally and functionally conserved residues are shown in stick. ELL2 is shown in light blue while Occludin is shown in cyan. All structural figures in this paper are made with Pymol.

followed by two α-helices to the CycT1 surface. Nuclear magnetic resonance studies showed that AFF4 residues 761–774 fold into a β-strand that combines with two strands of the AF9 AFF4-binding domain to generate a three-stranded β-sheet[18]. The structures of the P-TEFb and AF9 complexes with AFF4 revealed two of the three known interfaces used by AFF4 in assembly of the SEC. In this study, we set out to visualize the last of the three known interfaces critical for AFF4 function, its binding site for ELL1/2.

Progress in characterizing the AFF4 interface with ELL2 has been slower than for the P-TEFb and ENL/AF9 interfaces, in part because the AFF4-binding domain of ELL2 is poorly soluble and prone to aggregation. Here we work with a fusion construct such that a stable obligate complex between ELL2 and AFF4 is formed. This fusion-based tethered complex is stable and soluble enough to be crystallized. The crystal structure confirms that the AFF4-binding domain of ELL2 has an occludin fold, as predicted from the sequence homology[19]. It shows that the IDR consisting of AFF4 residues 301–351 (hereafter referred to as AFF4$^{ELLBow}$ for ELL1/2 binding) folds up into a helix and elbow joint arrangement that makes extensive contacts with the occludin domain of ELL2 (hereafter ELL2$^{Occ}$). These results complete the structural picture of how AFF1/4 engages its three known partners in the SEC.

## Results

**Mapping the AFF4$^{ELLBow}$ and ELL2$^{Occ}$ interaction.** Following the initial mapping of the AFF4 and ELL2 interaction sites to approximately residues 318–337 of the former and 519–640 of the latter[20] (Fig. 1a), we sought to isolate a stable form of this monomeric complex for crystallization (Supplementary Fig. 1a). It was difficult to obtain diffraction-quality crystals of ELL2$^{Occ}$ constructs with AFF4$^{ELLBow}$ fragments because of the propensity of the ELL2 fragment to aggregate over time. We reasoned that fusion of

AFF4$^{ELLBow}$ and ELL2$^{Occ}$ fragments might protect the AFF4-binding epitope on ELL2$^{Occ}$ from aggregation. Constructs were generated for both AFF4$^{ELLBow}$–(Gly-Ser)$_4$–ELL2$^{Occ}$ and ELL2$^{Occ}$–(Gly-Ser)$_4$–AFF4$^{ELLBow}$. The ELL2$^{Occ}$–(Gly-Ser)$_4$–AFF4$^{ELLBow}$ dimerized in solution, while AFF4$^{ELLBow}$–(Gly-Ser)$_4$–ELL2$^{Occ}$ was monomeric (Supplementary Fig. 1b). Given that the unfused fragments were monomeric, we concluded that the dimerization of ELL2$^{Occ}$–(Gly-Ser)$_4$–AFF4$^{ELLBow}$ represented a domain-swapping artifact (Supplementary Fig. 1c) and focused efforts on AFF4$^{ELLBow}$–(Gly-Ser)$_4$–ELL2$^{Occ}$.

**Structure of the AFF4$^{ELLBow}$ complex with ELL2$^{Occ}$.** The structure of the AFF4$^{ELLBow}$–ELL2$^{Occ}$ complex was determined by Selenomethionyl (SeMet) multiwavelength anomalous dispersion (MAD) phasing (Fig. 1b; Supplementary Fig. 2; Table 1). Helix α1 (residues 538–578) of ELL2 bends inward at Tyr552 by 30° such that the C-terminal part of α1 (553–578) packs against α2 (Fig. 1c). Helices α2 (584–602) and α3 (607–638) of ELL2 are oriented at an angle of ∼100° with respect to each other such that both pack along the length of the long, bent helix α1 (Fig. 1c). The structure confirms that ELL2$^{Occ}$ has a similar arch-shaped three-helix fold as the C-terminal domain of occludin[19]. The ELL2$^{Occ}$ and occludin C-terminal domain (pdb entry 1XAW) structures can be superimposed with an root mean squared deviation of 4.0 Å for 104 residue pairs (Fig. 1d). The main differences are in the α2–α3 connector and in the mutual orientation of these two helices. The α2–α3 angle is steeper in ELL2$^{Occ}$ than in occludin. A minor difference is that ELL2$^{Occ}$ has an extra single-turn helix, denoted α0, at its N terminus.

AFF4$^{ELLBow}$ is ordered over residues 314–349 and buries 1,535 Å$^2$ of solvent-accessible surface area. Fully 37% of the entire solvent-accessible surface area of AFF4$^{ELLBow}$ is buried in the contact. The AFF4$^{ELLBow}$ sequence folds into several distinct regions. It begins with helix α1 (315–324), is followed by

---

**Table 1 | Statistics of crystallographic data reduction and refinement.**

|  | Native | SeMet (Se peak) | SeMet (Se high remote) |
|---|---|---|---|
| *Data collection* |  |  |  |
| Space group | $P2_12_12_1$ | $P2_12_12_1$ | $P2_12_12_1$ |
| Unit cell parameters |  |  |  |
| *a, b, c* (Å) | 52.866, 57.324, 61.805 | 52.641, 57.422, 61.338 | 52.641, 57.422, 61.338 |
| *α, β, γ* (°) | 90.000, 90.000, 90.000 | 90.000, 90.000, 90.000 | 90.000, 90.000, 90.000 |
| Wavelength (Å) | 1.12709 | 0.9797 | 0.9569 |
| Resolution (Å) | 50.00–2.51 (2.60–2.51) | 50.000–2.10 (2.14–2.10) | 50.00–2.10 (2.18–2.10) |
| No. of reflections | 36,437 | 159,481 | 159,642 |
| Completeness (%) | 99.1 (92.3) | 100.0 (100.0) | 100 (100) |
| Redundancy | 5.4 (4.0) | 14.1 (13.9) | 14.1 (14.2) |
| Rsym | 0.139 (0.917) | 0.748(0.141) | 0.132 (0.72) |
| $<I>/<\sigma(I)>$ | 10.83 (1.05) | 33.7 (3.71) | 20.59 (4.06) |
| $CC_{1/2}$ | 0.701 | 0.919 | 0.925 |
|  |  |  |  |
| *Refinement* |  |  |  |
| Resolution (Å) |  | 41.92–2.003(2.075–2.003) |  |
| $R_{work}/R_{free}$ (%) |  | 19.71/24.83(28.04/40.22) |  |
| Average *B*-factor (Å) |  | 40.79 |  |
| R.m.s.d from ideality |  |  |  |
| Bond length (Å) |  | 0.003 |  |
| Bond angle (°) |  | 0.57 |  |
| *Ramachandran plot (%)* |  |  |  |
| Favoured |  | 98 |  |
| Allowed |  | 2 |  |
| Outliers |  | 0 |  |

R.m.s.d, root mean squared deviation.
Values in parentheses are for the highest-resolution shell. $R_{free}$ was calculated with 5% of the reflections selected in the thin shell.

---

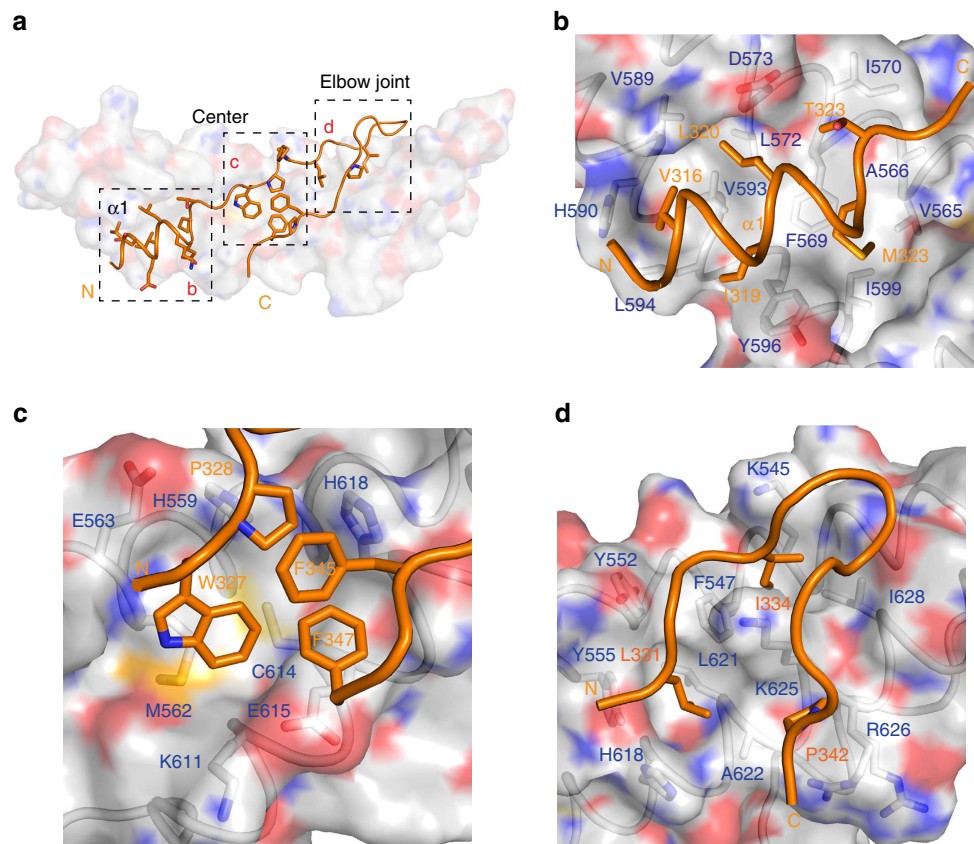

**Figure 2 | AFF4ELLBow–ELL2Occ interaction surfaces.** (**a**) Overview of the main binding determinants of the AFF4 ELLBow. (**b**) The first helix of the ELLBow (orange) binds in a hydrophobic groove on ELL2 (grey). The key residues are shown in a stick model. (**c**) The central cluster, in which hydrophobic residues of the ELLBow pack against ELL2 and one another, and are supplemented by polar interactions. Water molecules are shown as red spheres. (**d**) The ELLBow joint.

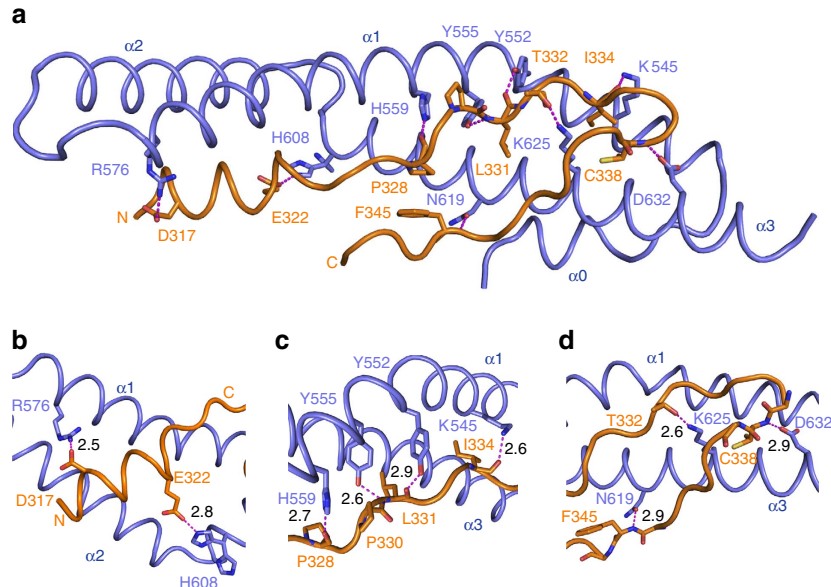

**Figure 3 | Hydrogen bonding in the AFF4ELLBow–ELL2Occ complex.** (**a**) Overview of the network of hydrogen bonds. The residues involved in the hydrogen bonding are shown in stick. Hydrogen bonds are shown as magenta-coloured dashed lines. (**b–d**) Details of the hydrogen bonds. The length of the hydrogen bonds is indicated next to the dashed lines.

an extended hydrophobic sequence (325–327), a polyproline segment (328–330), an extended region that doubles back on itself in what we refer to as the ELLBow joint (331–343), and a second extended hydrophobic sequence (344–349) (Fig. 2a). The fusion construct contains 17 residues that are not visualized in electron density. These include AFF4 350–351,

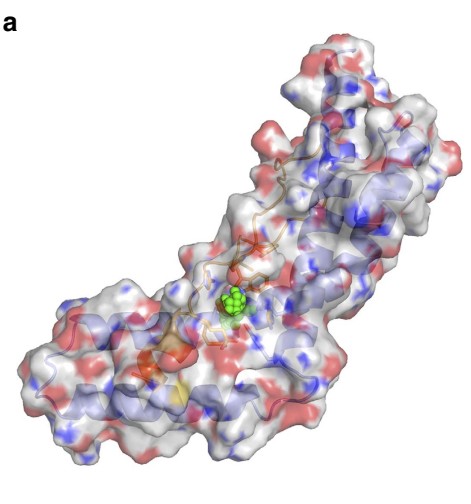

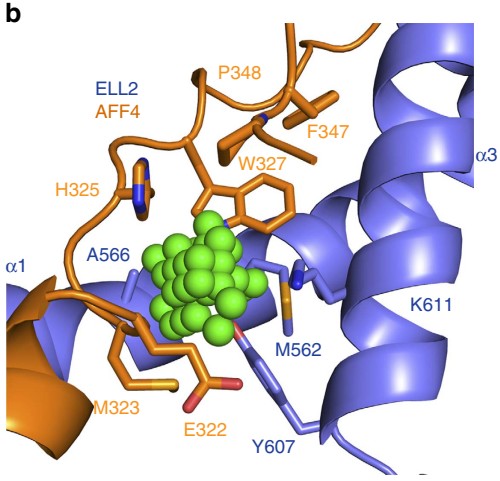

**Figure 4 | Cavity at the AFF4$^{ELLBow}$–ELL2$^{Occ}$ interface.** (**a**) Overall view of the cavity that might work as a potential drug-binding site. Molecules of AFF4$^{ELLBow}$:ELL2$^{Occ}$ complex are shown in translucent cartoon and surface model colours are light blue, ELL2; orange, AFF4; and green, dummy atoms (**b**) Close-up of the cavity region as represented and coloured in **a**.

followed by 8 Gly-Ser linker residues and ELL2 residues 519–524. These 17 residues are more than adequate to span the 15 Å gap between the carbonyl carbon of AFF4 residue 349 and the amide nitrogen of ELL2 residue 525 in the structure. Hydrophobic side chains of AFF4$^{ELLBow}$ α1, including Val316, Ile319, Leu320 and Met323, are buried in a hydrophobic groove formed by the C-terminal half of ELL2$^{Occ}$ α1 and α2 (Fig. 2b). These helices of ELL2$^{Occ}$ contribute residues Val565, Phe569, Ile570, Leu572, Asp573, Val589, His590, Tyr596, Leu594 and Ile599 to the AFF4 α1-binding site (Fig. 2b). ELL2$^{Occ}$ buries 1,315 Å$^2$ of solvent-accessible surface area, corresponding to 15% of its total surface area.

AFF4$^{ELLBow}$ is centred on Trp327, which forms extensive hydrophobic interactions with the side chains of ELL2 residues His559, Met562, Cys614 and Glu615. The Trp327 indole nitrogen also forms a water-mediated hydrogen bond with the Tyr607 hydroxyl. This cluster of residues is completed by the side chains of AFF4 Pro328, Phe345 and Phe347 (Fig. 2c). Collectively, this cluster forms an extensive interaction network, in which AFF4$^{ELLBow}$ folds up not only against ELL2 but also against itself.

In the AFF4$^{ELLBow}$ joint, the side chain of Leu331 sticks into a pocket comprising Tyr552, Tyr555, His618, Leu621 and Ala622

of the N-terminal half of ELL2$^{Occ}$ α1 and α3. The side chain of Ile334 packs against the side chains of Lys545, Phe547, Lys625 and Leu628. At the distal end of ELLBow joint, Pro342 falls into a shallow cavity composed of Ala622, Lys625 and Arg626 (Fig. 2d).

A number of hydrogen bonds are observed in the complex (Fig. 3a). In AFF4$^{ELLBow}$ α1, the side chains of Asp317 and Arg576 of ELL2 form a bidentate salt bridge with one another (Fig. 3b). Glu322 of AFF4 forms a 2.8 Å salt bridge with one of the two observed rotamers of His608 of ELL2 (Fig. 3b). In the central cluster, the carbonyl group of Pro328 forms a 2.7 Å hydrogen bond with the side chain of His559 of ELL2 (Fig. 3c). Moving into the ELLBow joint, the main-chain amide and carbonyl of AFF4 Leu331 form hydrogen bonds with the hydroxyl oxygens of Tyr552 and Tyr555, respectively, of ELL2. A 2.6 Å hydrogen bond is formed between Thr332 of AFF4 and Lys625 of ELL2 (Fig. 3d). The Ile334 carbonyl accepts a hydrogen bond from the side chain of Lys545. The Cys338 main-chain amide donates a hydrogen bond to the side chain of Asp632. The main-chain amide of Phe345 forms a 2.9 Å hydrogen bond with the side chain of Gln619 (Fig. 3d).

The AFF4$^{ELLBow}$–ELL2$^{Occ}$ complex was screened for cavities using POCASA 1.1 (POcket-CAvity Search Application)[21] with a probe radius of 3 Å. Of the five largest cavities located, one is an internal cavity at the AFF4$^{ELLBow}$–ELL2$^{Occ}$ interface (Fig. 4a). The cavity is 36 Å$^3$ in volume and is connected to the exterior by a narrow mouth (Fig. 4a). It is lined by the aliphatic part of Glu322, Met323, His325, Trp327, Phe347 and Pro348 of AFF4, and by Met562, Ala566, Tyr607 and the aliphatic part of Lys611 of ELL2 (Fig. 4b). These residues are in or adjoin the central cluster part of the interface.

**Function of the AFF4$^{ELLBow}$ interface with ELL2$^{Occ}$ in binding.** To validate whether the observed structural interface corresponded to the mode of binding of AFF4$^{ELLBow}$ and ELL2$^{Occ}$ in solution, we carried out a series of mutant peptide binding assays using fluorescence polarization. We considered this particularly critical given the use of the fusion construct to obtain crystals. The assay monitored the displacement of fluorescently labelled wild-type AFF1$^{ELLBow}$ peptide by unlabelled mutant peptides 301–351. The unlabelled wild-type peptide in this system has $K_d = 86$ nM (Supplementary Table 1; Fig. 5a). The AFF4 hydrophobic residues Val316, Ile319, Leu320, Met323, Trp327, Leu331, Ile334 and Pro342, were mutated to Asp to maximally destabilize hydrophobic interactions. Consistent with expectation, mutation of multiple hydrophobic residues to Asp resulted in large decreases in affinity. The double mutant I319D/L320D reduced affinity by >25-fold (Supplementary Table 1; Fig. 5a). The $K_d$ for the triple mutant I319D/L320D/M323D was immeasurable due to weak binding, but >3 µM, representing a ~50-fold loss of affinity (Supplementary Table 1; Fig. 5a). The same was true of two other triple hydrophobic mutants tested, M323D/L331D/I334D and W327D/L331D/I334D (Supplementary Table 1; Fig. 5b). The single mutant M323D has the largest effect of any single-amino-acid change, with a reduction in affinity of >25-fold (Supplementary Table 1; Fig. 5b). Moving closer to the centre of the AFF4$^{ELLBow}$, L331D and I334D reduce affinity by ~20- and 8-fold, respectively (Supplementary Table 1; Fig. 5b). This highlights the role of hydrophobic residues in AFF4$^{ELLBow}$ helix α1 and immediately C terminal to it in the central cluster, as the critical anchor points and affinity determinants.

Hydrophobic residues of the central cluster make smaller contributions than those highlighted above. W327D reduces affinity fourfold, while F345D/F347D reduces it by less than

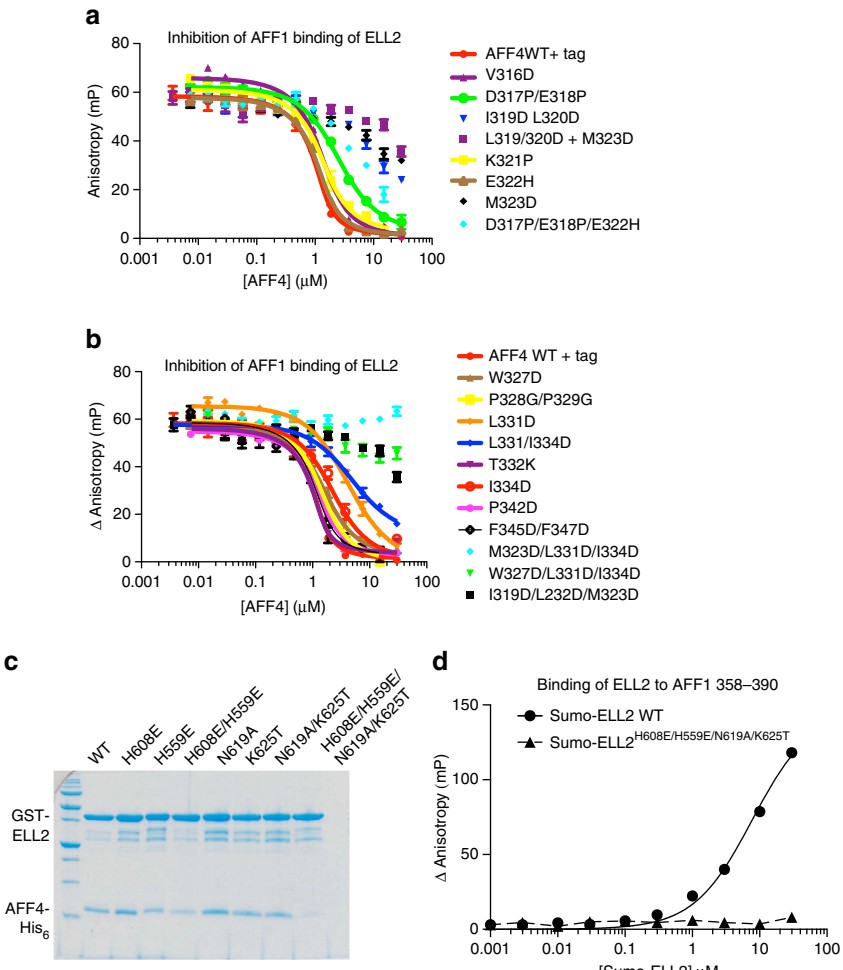

**Figure 5 | Contributions of AFF4 ELLBow interactions to binding in solution.** (**a**) Binding of AFF4[ELLBow] wild type (WT) and mutants in the N-terminal α-helix to Sumo-ELL2[Occ]. Sumo-ELL2[Occ] binding to fluorescently labelled AFF1[ELLBow] peptide 358–390 is competitively inhibited by increasing amounts of AFF4[ELLBow], as described in experimental procedures. Error bars reflect the s.e. from three experimental replicates in **a**,**b**. (**b**) Binding of AFF4[ELLBow] WT and mutants in the central cluster and elbow joint, assayed as in **a**. (**c**) GST fusions of the indicated ELL2[Occ] mutants were immobilized and their ability to pull-down His6-tagged WT AFF4[ELLBow] assessed. An uncropped version of this pull-down gel is shown in Supplementary Fig. 4. (**d**) Direct binding of WT and mutant Sumo-ELL2 to fluorescently labelled AFF1[ELLBow] peptide. The assay was performed in triplicate. The s.e. from the three replicates in **d** is smaller than the symbols used to plot the data points.

twofold. P342D led to a similar threefold drop (Supplementary Table 1; Fig. 5b). These more modest contributions may reflect that these side chains are partially solvent accessible in the AFF4[ELLBow]–ELL2[Occ] complex. Moreover, their interactions are made in part with other residues within the AFF4[ELLBow] such that they could potentially make residual hydrophobic interactions even in unbound AFF4. The polyproline helix does not seem to have a major role in affinity, with the double 328–329 Pro–Gly mutant reducing affinity only by a factor of three (Supplementary Table 1; Fig. 5b).

The interface has a significant polar component, with some hydrophilic residues contributing substantially to binding, and others less so. The AFF4[ELLBow] α1 mutant D317P/E317P was designed to disrupt hydrogen bonding, involving Asp317 and to introduce helix breaker mutants in α1. This mutation lowered affinity by 10-fold (Supplementary Table 1; Fig. 5a). The charge reversal mutation E322H reduced affinity by less than twofold (Supplementary Table 1; Fig. 5a).

It proved impossible to purify hydrophobic to Asp mutants in the AFF4-binding site of ELL2[Occ] because these proteins were insoluble when expressed in *Escherichia coli*. Presumably,

this is because these hydrophobic residues also contribute to the hydrophobic core of the ELL2[Occ] fold. It was, however, possible to purify ELL2[Occ] polar mutants in the binding site. We examined the roles of ELL2 His559, His608, Asn619 and Lys625 by pull-down assay (Fig. 5c). Single mutants H559E, H608E, N619A and K625T had no apparent effect on binding by pull-down. However, the quadruple mutant H559E/H608E/N619A/K625T completely abrogated binding both in the pull-down assay (Fig. 5c) and in a fluorescence polarization binding assay (Fig. 5d). This validates the role of these residues in the interface in solution.

**The AFF4[ELLBow] and ELL2[Occ] interface *in vivo*.** It had previously been shown that the AFF4 sequence 318–337 was sufficient for ELL2 binding[20] and that AFF4 can heterooligomerize AFF1 via its C-terminal domain[22]. To prevent the endogenous AFF1 from rescuing the mutant construct, function was tested in the context of a deletion of the C-terminal sequence 970–1,163. Double deletion of AFF4 residues 318–337 and 970–1,163 abrogated the interaction

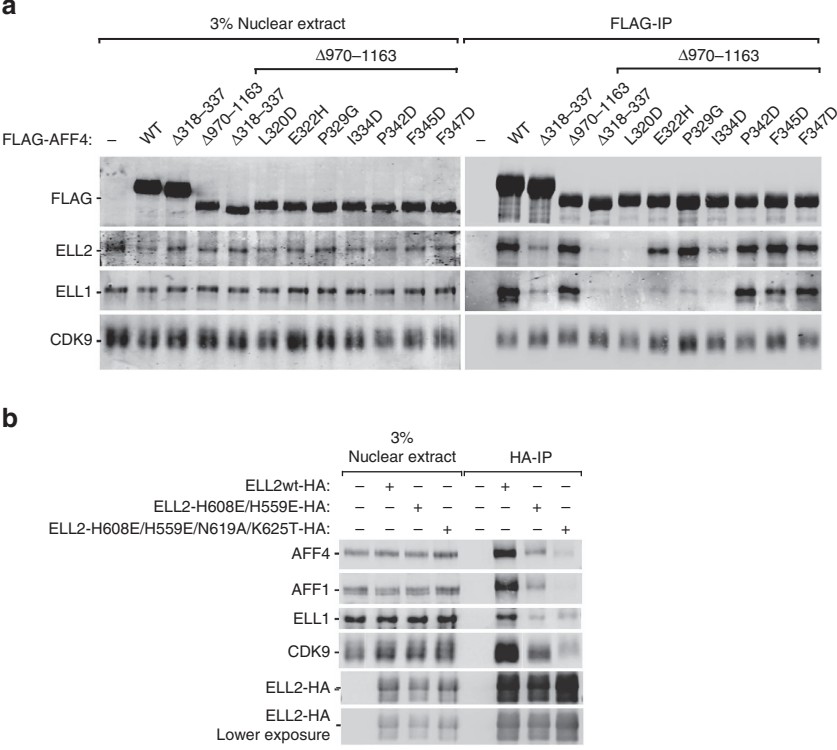

**Figure 6 | Role of the ELLBow in AFF4 interactions with ELL1/2 in nuclear extracts.** (**a,b**) Nuclear extracts (NE) were prepared from HEK 293T cells transfected with the indicated plasmids and subjected to immunoprecipitation (IP) with anti-FLAG (**a**) or anti-HA (**b**) agarose beads. The NE inputs and IP eluates were examined by immunoblotting for the presence of the various proteins indicated on the left. A shorter exposure of the ELL2-HA panel is also shown at the bottom of **b**. Uncropped versions of the blots are shown in Supplementary Fig. 5.

between AFF4 and ELL1/2 completely (Fig. 6a). To determine if single residues within AFF4$^{ELLBow}$ contributed to binding and function in cells, point mutants were constructed in the context of AFF4 Δ970–1,163. ELL1 contains a C-terminal domain homologous to that of ELL2, hence binding to ELL1 was also tested. L320D was most effective, blocking both ELL1 and ELL2, consistent with its very strong effect on binding *in vitro* (Fig. 6a; Supplementary Table 1). E322H, P329G and I334D partially blocked ELL2 binding, but completely knocked out ELL1 binding, consistent with their intermediate effects on *in vitro* peptide binding. Both ELL1 and ELL2 bound robustly to the mutants P324D, F345D and F347D, consistent with their two- to threefold effects on binding *in vitro* (Fig. 6a; Supplementary Table 1).

To determine if the AFF4-binding site on ELL2 was functional in cells, polar mutants were inserted into ELL2 alleles and these were transfected into HeLa cells.

We avoided testing hydrophobic mutants of ELL2, since we had previously found that these destabilized the ELL2 structure. HA-tagged ELL2$^{H559E/H608E}$ and ELL2$^{H559E/H608E/N619A/K625T}$ were expressed at essentially wild-type levels in HeLa cells (Fig. 6b). Wild-type HA-ELL2 pulled down AFF1, AFF4 and ELL1 from extracts. ELL2$^{H559E/H608E}$ has sharply reduced binding to AFF1, AFF4 and ELL1. ELL2$^{H559E/H608E/N619A/K625T}$ has only trace binding to AFF1 and AFF4 in extracts. These findings support that the structural interface is responsible for the interaction of ELL2 with both AFF1 and AFF4 in cells.

**Role of the interface in proviral transactivation.** Overexpression of AFF4 stimulates proviral transcription by ∼5–9-fold and ∼26-fold in HEK 293 T and HeLa cells, respectively (Fig. 7a).

Deletion of the C-terminal ELL1/2-binding domain almost completely blocked transactivation (Fig. 7a). The residual activity of AFF4 Δ970–1,163 was so low that meaningful results could not be obtained for transactivation phenotypes of these mutants (Fig. 7a). The abundance of the SEC complex appears to be limiting for transactivation such that overexpression of ELL2 in the presence of extra AFF4 promotes transcription by a factor of 14 (Fig. 7b). Polar mutants in the AFF4-binding site of ELL2$^{Occ}$ were tested for their effects on transcription. ELL2$^{H559E/H608E}$ and ELL2$^{H559E/H608E/N619A/K625T}$ had threefold and fivefold less transactivation activity, respectively, than wild type. Very similar three- to fourfold effects are seen in Jurkat 2D10 cells (Fig. 7c). These observations strongly support a functional role for the AFF4$^{ELLBow}$-binding site on ELL2$^{Occ}$ in transactivation.

**Discussion**

The crystallization of the AFF4$^{ELLBow}$–ELL2$^{Occ}$ complex rounds out our structural-level understanding of how the AFF4 scaffold recruits its three known partners in the SEC, P-TEFb, ENL/AF9 and ELL1/2. The limited solubility of ELL2$^{Occ}$ made this a more challenging target for crystallization, hence the necessity for the fusion approach. When using protein chimeras as a basis for structure solution, it is particularly critical to validate the findings in solution and in functional assays. One area that remains to be further explored is the relationship between ELL1/2 binding to AFF1/4 and the putative hetero-dimerization mediated by the C-terminal domains of AFF1/4. Binding assays *in vitro*, pull-downs from nuclear extracts, and proviral transactivation assays present a unified, consistent picture that validates the structural results.

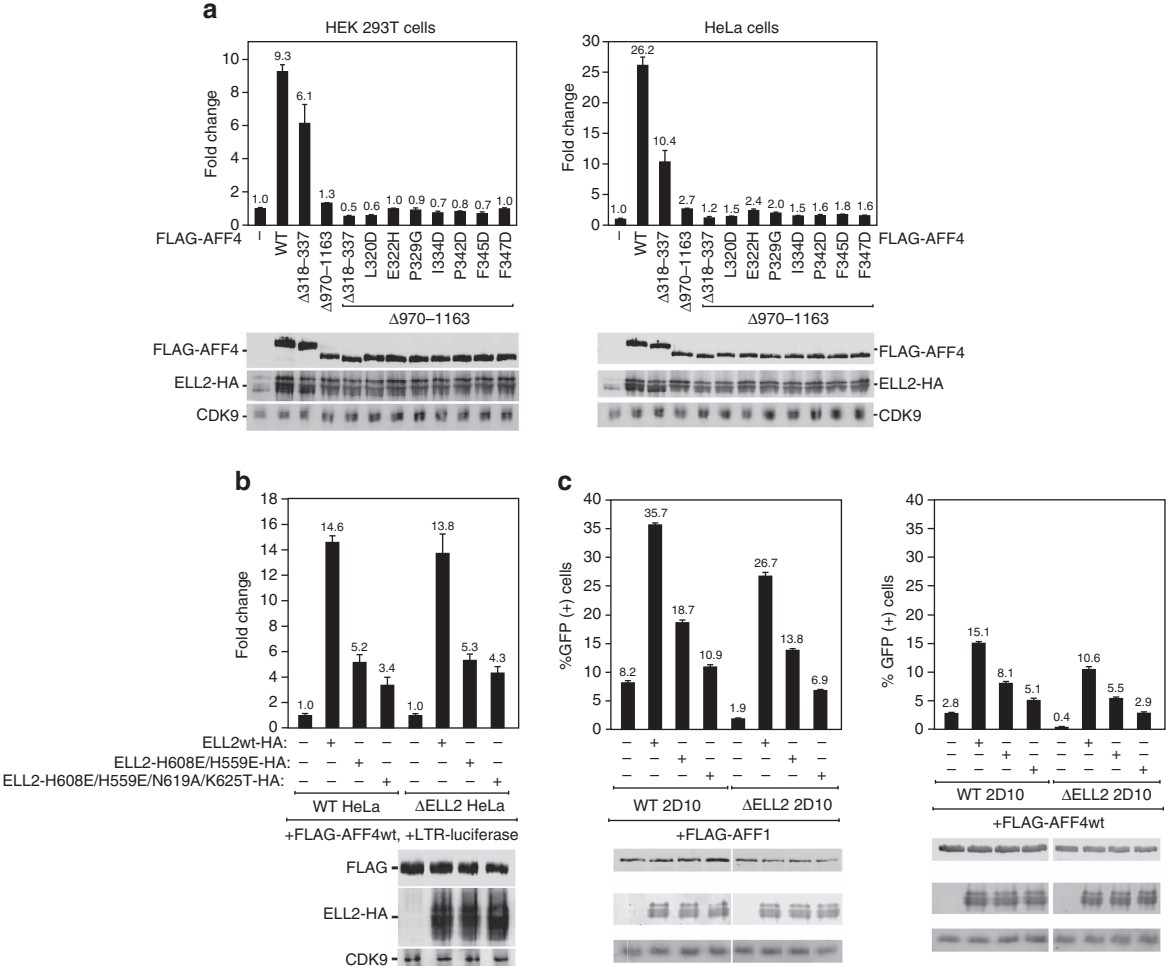

**Figure 7 | The ELLBow-binding site of ELL2 is important for HIV-1 long terminal repeat (LTR) transcription.** (**a,b**) Luciferase activities were measured and analysed in extracts of cells transfected in triplicate with the HIV-1 LTR-luciferase construct together with the combinations of plasmids expressing wild-type and mutant ELL2-HA and FLAG-AFF4 as indicated. Each of the ELL2 and AFF4 plasmids was transfected at 1 μg per well. The activities in the control groups were set to 1. The error bars represent mean ± s.d. from triplicate wells. An aliquot of each cell extract was examined by immunoblotting for presence of the various proteins labelled on the left. (**c**) Jurkat 2D10 cells were nucleofected in triplicates with plasmids expressing AFF1 (top) or AFF4 (bottom) alone or in combination with ELL2wt or its mutants as indicated. 48 h post nucleofection, the percentages of GFP + cells were measured by flow cytometry and plotted. The error bars represent mean ± s.d.'s. The levels of the indicated proteins in nucleofected cells were determined by immunoblotting. Uncropped versions of the blots are shown in Supplementary Fig. 6.

The structure confirms the decade-old prediction that the C-terminal domains of ELL1/2 would have the same fold as the occludin ZO-1-binding domain. Occludin is a transmembrane tight junction protein that has no known involvement in transcription. It is not clear why this protein and ELL1/2 should share a domain uniquely present in this small set of otherwise unrelated proteins. In the initial analysis of the occludin structure, it was proposed that another tight junction protein, ZO-1, bound to a basic patch at the concave center of the arch[19]. This patch of occludin includes Lys504 and Lys511, which correspond structurally to the functionally important His618 and Lys625 in the AFF1/4-binding site of ELL2 (Fig. 1e). Subsequently, another report proposed that ZO-1 bound elsewhere, at one tip of the occludin domain arch. Despite these uncertainties, the structural similarities are extensive enough to suggest a common evolutionary origin and related protein-binding functions for the three-helical domains of occludin and ELL1/2.

The bromodomain and extraterminal protein inhibitor JQ1 (ref. 23) and related latency-reversing agents promote reactivation of HIV-1 from latency via P-TEFb[24–28]. New classes of HIV-1 latency-reversing agents are being sought in the context of HIV eradication strategies[2]. We observed a cavity at the AFF4–ELL2 interface that appears likely to be present also in the AFF1–ELL2 complex relevant to proviral activation[14], on the basis of the complete identity of the AFF1 and AFF4 residues involved. If so, this could provide an avenue for the design of new SEC activators with JQ1-like effects on latency, but acting by an orthogonal molecular mechanism.

## Methods

**Cloning and protein purification.** DNAs for ELL2 fragments and AFF4–ELL2 fusions were subcloned into pGST-parallel2, and DNAs for AFF4 peptide fragments were subcloned into pRSFduet-1 and pHis-parallel2. Plasmids expressing FLAG-tagged wild-type AFF4 and HA-tagged wild-type ELL2 were generated using the primers described in Supplementary Table 2 (ref. 5). The plasmids expressing mutant versions of AFF4 and ELL2 were generated by PCR mutagenesis. The mutant constructs were verified by DNA sequencing. All proteins were expressed in *E. coli* BL21-gold (DE3) cells (Agilent Technologies). After induction with 0.2 mM isopropyl-β-D-thiogalactoside overnight at 16 °C, the cells were pelleted by centrifugation at 4,000*g* for 10 min. Cell pellets were lysed in

25 mM Tris-HCl pH 8.0, 150 mM NaCl, 0.5 mM TCEP-HCl and 1 mM phenylmethylsulphonyl fluoride by ultrasonication. The lysate were centrifuged at 25,000$g$ for 1 h at 4 °C. The supernatants for ELL2 and its fusions were loaded onto GS4B resin at 4 °C, target proteins were eluted and the eluate applied to a Hi Trap Q HP column. Peak fractions were collected and digested with tobacco etch virus protease at 4 °C overnight. Tobacco etch virus and GST were removed by loading the solution onto Ni-NTA and GS4B columns, respectively. Target proteins were further purified on a Superdex 200 16/60 column equilibrated with 25 mM Tris-HCl pH 8.0, 150 mM NaCl and 0.5 mM TCEP-HCl. The peak fractions were collected and flash-frozen in liquid $N_2$ for storage. The supernatant of AFF4 was loaded onto Ni-NTA resin at 4 °C, eluted with an imidazole gradient, and applied to a Superdex 75 16/60 column equilibrated with 25 mM Tris-HCl pH 8.0, 150 mM NaCl, 0.5 mM TCEP-HCl. SeMet protein was expressed in *E. coli* BL21-gold (DE3) cells grown in M9 minimal medium supplemented with 5% LB medium. An amount of 0.2 mM isopropyl-β-D-thiogalactoside and 100 mg selenomethionine were added when the $OD_{600}$ reached 1.0. Cells were pelleted by centrifugation at 4,000$g$ for 10 min after overnight induction at 16 °C. SeMet AFF4$^{301-351}$–(Gly-Ser)$_4$–ELL2$^{519-640}$ was prepared as above and SeMet incorporation verified by mass spectrometry.

**Crystallization of the AFF4$^{ELLBow}$–ELL2$^{Occ}$ fusion.** The purified fusion construct AFF4 (301–351)–(Gly-Ser)$_4$–ELL2 (519–640) was concentrated to 10 mg ml$^{-1}$ with a 10 kD centrifugal filter (Millipore). Crystals were grown by hanging-drop vapour diffusion at 19 °C. The protein solution was mixed with well buffer composed of 0.2 M NaCl, 10 mM $MgCl_2$, 0.3 M $Na_3$ citrate, 0.2 M Na thiocyanate and 0.1 M Hepes pH 7.4. Crystals appeared in 24 h and grew to full size in 5 days. Crystals were flash-frozen with liquid $N_2$ in well buffer. SeMet crystals were grown in the same condition as native crystals. Native data were collected on BL7-1 at Stanford Synchrotron Radiation Lightsource. Native crystals diffracted to 2.5 Å and data were collected at a wavelength of 1.1271 Å. The structure was solved using data from SeMet crystals as described in the main text.

**Pull-down assays.** Mutants of ELL2 (519–640) and AFF4 (300–351) were purified as described above. The concentration of proteins and peptides was determined by ultraviolet absorption at 260–280 nm. A measure of 9 μM GST-ELL2 and 20 μM His$_6$-AFF4 were incubated with GS4B resin at 4 °C for 2 h in 80 μl of 25 mM Tris-HCl pH 8.0, 150 mM NaCl and 0.5 mM TCEP-HCl. The resin was washed three times with the incubation buffer. Then, the resin was boiled in 30 μl 1 × SDS loading buffer at 95 °C for 5 min before being applied to SDS–polyacrylamide gel electrophoresis for analysis.

**Fluorescence polarization.** Protein binding was measured using the fluorescence anisotropy of a 33-residue segment of AFF1 (residues 358–390), following procedures similar to those used previously to characterize AFF4 binding to P-TEFb[17]. AFF1 358–390 are almost identical to AFF4 318–350 with only three amino-acid changes between the two homologues. The AFF1 peptide C-FAM-GABA-EILKEMTHSWPPPLTAIHTPSTAEPSKFPFPTK-amide was synthesized (University of Utah DNA/Peptide Facility), where FAM indicates 5-carboxyfluoroscein and GABA indicates a γ-amino-butyric acid spacer. Competition titration experiments with unlabeled His-tagged AFF4 protein 301–351 were performed using 2 μM Sumo-ELL2 in 25 mM HEPES pH 7.5, 100 mM NaCl, 10% glycerol, 0.05% NP-40, 0.5 mM TCEP and 5 nM fluorescent peptide. A Victor 3V (Perkin Elmer) multi-label plate reader was used to measure fluorescence anisotropy. Three experimental replicates were carried out for each curve. Binding curves were fit to a formula describing competitive binding of two different ligands to a protein[29] using Prism version 5.0c (Graphpad Software).

**Co-immunoprecipitation.** Approximately 2 × 10$^7$ HEK 293T cells (UC Berkeley Cell Culture facility) in two 145-mm dishes were transfected by plasmids expressing the wild-type or mutant FLAG-AFF4 or ELL2-HA (20 μg each). Forty-eight hour after transfection, the cells were harvested and swollen in 4 ml hypotonic buffer A (10 mM HEPES-KOH (pH 7.9), 1.5 mM $MgCl_2$ and 10 mM KCl) for 5 min and then centrifuged at 362$g$ for 5 min. The cells were then disrupted by grinding 20 times with a Dounce tissue homogenizer in 2 ml buffer A, followed by centrifugation at 3,220$g$ for 10 min to collect the nuclei. The nuclei were then extracted in 400 μl buffer C (20 mM HEPES-KOH (pH 7.9), 0.42 M NaCl, 25% glycerol, 0.2 mM EDTA, 1.5 mM $MgCl_2$, 0.4% NP-40, 1 mM dithiothreitol and 1 × protease inhibitor cocktail) on ice for 30 min, followed by centrifugation at 20,800$g$ for 30 min. The supernatant (nuclear extracts (NE)) was then mixed with 10 μl of anti-FLAG agarose (A2220 Sigma) or anti-HA agarose (A2095 Sigma) and rotated at 4 °C overnight. The beads were then washed three times with buffer D (20 mM HEPES-KOH (pH 7.9), 0.3 M KCl, 15% glycerol, 0.2 mM EDTA and 0.4% NP-40), and eluted with 30 μl 0.1 M glycine-HCl (pH 2.0). For western blot, 3% of the NE input and 50% of the immunoprecipitation eluate were loaded into each NE and immunoprecipitation lane, respectively. Primary antibodies used for western blots are: mouse anti-FLAG (F1804, Sigma), rat anti-HA (11867423001, Roche), rabbit anti-human ELL2 (A302–505A, Bethyl), rabbit anti-human ELL1 (A301–645A, Bethyl), rabbit anti-human AFF1

(A302–344A, Bethyl), mouse anti-human AFF4 (ab57077, Abcam) and mouse anti-human α-tubulin (CP06, EMD CHEMICALS). Secondary antibodies used for western blots are: goat anti-mouse-680 nm (A-21057, Invitrogen), goat anti-rabbit-680 nm (A-21076, Invitrogen) and goat anti-rat-800 nm (612-132-120, Rockland). For endogenous proteins except α-tubulin, the primary antibodies were diluted to 1 μg ml$^{-1}$, for FLAG/HA tags and α-tubulin, the primary antibodies were diluted 5,000-fold. Secondary antibodies were diluted 10,000-fold.

**Luciferase reporter assay.** Approximately 6 × 10$^5$ HEK 293T cells or 4 × 10$^5$ HeLa cells (UC Berkeley Cell Culture facility) in six-well plates were transfected in triplicate by plasmids expressing FLAG-AFF4 and/or ELL2-HA (1 μg each) with the HIV-1 LTR-luciferase construct (0.1 μg). Forty-eight hours after transfection, the cells were collected and lysed in 1 × reporter lysis buffer (E3971 Promega), followed by centrifugation at 20,800$g$ for 1 min. Luciferase activities in the supernatant were measured using the Luciferase Assay System (E1501 Promega) on a Lumat LB 9501 luminometer.

**CRISPR/Cas9-mediated knockout of ELL2 gene in HeLa cells.** The procedures and single-guide RNA sequence for generating the HeLa-based ELL2-KO knockout (KO) cell line dELL2 (ref. 14) were as follows. Briefly, forward (5′-CACCGAGCGCCCGGATCGCCGTCT-3′) and reverse(5′-AAACAGA CGGCGATCCGGGCGCTC-3′) DNA oligos containing the single-guide RNA sequence targeting the first exon of ELL2 were synthesized, annealed and cloned into the pSpCas9(BB)-2A-Puro vector (Addgene plasmid ID: 48,139), and transfected into HeLa cells, which were then selected by puromycin for 2 days, and diluted to single clones. The KO clone was initially identified by anti-ELL2 immunoblotting (Supplementary Fig. 3), and then verified by Sanger sequencing of the targeted genomic site.

**Test of the effects of ELL2 mutants on HIV latency reversal.** A total of 2 μg plasmids expressing AFF1/4 alone or in combination with ELL2wt or its mutants were nucleofected into 1 × 10$^6$ Jurkat 2D10 cells (gift of J. Karn, Case Western Reserve University)[30] using Amaxa kit V and the manufacture's protocols (X-005). GFP + cells indicating the reversal of HIV latency were measured by flow cytometry 48 h post nucleofection. Three biological repeats were done for each group, with their percentages of GFP + averaged and s.d.'s calculated to generate the error bars. An aliquot of cells from each group were lysed for immunoblotting with the indicated antibodies.

**Data availability.** Coordinates and structure factor of the structure reported here have been deposited into the Protein Data Bank with PDB Code: 5JW9. All additional experimental data are available from the corresponding author on reasonable request. The PDB Code 1XAW, UniProt accession codes Q9UHB7 and O00472 were used in this study.

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

## Acknowledgements

We thank Xuefeng Ren, James Holton and George Meigs for assistance with data collection and Bo Wan for the construction of the HeLa-based ELL2-KO cell line. This work was supported by NIH grants P50GM082250 (J.H.H.), and NIAID R01AI041757 and R01AI095057 (Q.Z.), and NSFC grant 81671388 (Q.S.). The Minstrel crystal farm was purchased with support from the NIH, S10 OD016268. Beamline 8.3.1 at the Advanced Light Source, LBNL, is supported by the U. C. Office of the President, Multicampus Research Programs and Initiatives grant MR-15-328599 and the Program for Breakthrough Biomedical Research, which is partially funded by the Sandler Foundation. The Advanced Light Source is supported by the Director, Office of Science, Office of Basic Energy Sciences, of the U.S. Department of Energy under Contract No. DE-AC02-05CH11231. The Stanford Synchrotron Radiation Lightsource is supported by the U.S.D.O.E. under contract No. DE-AC02-76SF00515. The SSRL Structural Molecular Biology Program is supported by the DOE Office of Biological and Environmental Research and by NIH grant P41GM103393.

## Author contributions

S.Q. performed the structural biological study. Z.L. performed the cell biological experiments. U.S.-G. performed the fluorescence polarization assay. G.S. performed hydrogen-deuterium exchange coupled to mass spectrometry (HDX MS) analysis used to optimize crystallization constructs. J.H.H. and S.Q. wrote the manuscript. All the authors discussed the results and commented on the manuscript.

## Additional information

**Competing financial interests:** The authors declare no competing financial interests.

