## [Peer review file · Nature Communications]

Reviewers' comments:

Reviewer #1 (Remarks to the Author):

Qi and colleagues present a detailed study describing the crystal structure of the ELL1/2 and AFF4 binding surface using AFF4/ELL2 fusion proteins. This is followed by systematic point mutagenesis of AFF4 and ELL2 and in vitro and in vivo binding and functional studies using the HIV LTR as a model system. This is overall an interesting study that follows previous structural work of AFF4 and other SEC components. However, despite considerable and detailed work on the mutants, the overall functional relevance of the described interaction interface seems low as point mutants only show interaction and functional defects when a second not very well described binding site in AFF4 is also deleted. In fact, in Figure 6a, deletion of this new, not well-described, second site nearly completely abrogates transactivation of the HIV promoter by the SEC while deletion of the here-characterized site has only a minor effect. In addition, the study is framed to be significant for HIV latency reversal and activation of the SEC but this is not supported by the current data. All data (using mutants) are geared towards disruption of binding, and it is unclear how activation can be achieved. However, this is prominently featured in the title, abstract and throughout the text and should be toned down. In the abstract, the authors highlight a binding pocket for a small molecule in the crystal structure but the only figure showing the relevance of this pocket is Supplemental figure S4, making it a minor feature of the manuscript in its current form.

Specific Comments

Figure 1: There is a "d" over the crystal structure in figure 1C. Figure 1E is not discussed in the text of the results section.

Figure 4: Please explain why the ELL2-Occ peptide was sumolyated. 4A: Y-axis needs to be labeled. Please comment on biological replicates in the Figure legend.

Figure 5:

-Here it is unclear why the 970-1163 mutant is introduced in order to assess the binding, please comment more on how this domain was discovered and how binding of the two interfaces is envisioned.

-Why do E322H and P329G mutants exhibit decreased binding to ELL1 relative to ELL2? These mutants do not appear to have an effect in the FP experiments. What is special about ELL1 that confers differential binding? Please comment.

-5B: The authors should use a lighter exposure for the ELL2-HA blots. There appears to be less ELL2-H608E/H559E-HA mutant. Why is the binding of CDK9 lost? Is it through loss of AFF4 binding? Please comment. Please also comment on biological replicates.

Figure 6:

Figure 6a shows basically that the 970-1163 mutation is more important for HIV-1 transcription than the 318-337 mutation or any of the point mutations herein (Supplemental Figure). The testing of the point mutants in the context of the 970-1163 mutation does not contribute anything because the 970-1163 mutation alone already decreases activation to almost baseline levels. It is unclear how these data support the main message of the paper.

Figure 6b has no figure legend and the part of the figure labeled "dELL2" is not described in the text. As this is the only data that connect the structural findings with function via ELL2 it is critical to show the same experiments with AFF4 delta318-337 and AFF4 delta970-1163 and both to show that the ELL2 mutations decrease HIV transcription via interaction with AFF4318-337.

Reviewer #2 (Remarks to the Author):

The manuscript by Hurley and colleagues reports a straightforward structure-function analysis of the interaction between the transcription elongation factor ELL 1/2 and the scaffolding protein AFF 1/4. These proteins are components of the super elongation complex that transactivates HIV transcription upon recruitment by the tat-TAR complex in conjunction with P-TEFb. AFF1 (or 4) are intrinsically disordered proteins that interact with ELL 1/ 2 in the SEC. The N-terminal part of AFF4 has been determined in complex with Tat and most of P-TEFb. This manuscript adds to the description of the SEC by adding the interface of AFF4 with ELL2 to the already known interactions with P-TEFb and ENL. This was achieved by cleverly tethering the two proteins together to avoid aggregation of AFF 4. The most surprising observation is the extent to which the complex buries protein surface area. It further provides biochemical and functional (through transactivation assays) validation of the structural work and of how ELL is recruited to the SEC by traditional structure-guided mutagenesis.

The experimental work is well done and presented. The only thing I found unclear in this otherwise well-written and well-executed study is the mention in the abstract of a binding site for a small molecule at the interface, followed up briefly on page 8 and a supplementary figure. This seems a relevant information both for understanding modulation of the SEC by metabolites and potentially for pharmaceutical targeting, but it is only very briefly brushed upon in the main text, and could be expanded. Otherwise, it might easily be missed by readers.

Reviewer #3 (Remarks to the Author):

This is a very nice manuscript, which reports crystal structure of the ELL2 C-terminal domain bound to its cognate AFF4 fragment. Furthermore, the authors demonstrate the functional significance of the ELL2-AFF4 interacting interface for HIV-1 Tat-mediated proviral transcription. However, HDX studies are rather incomplete. The authors carried out one set of experiments using a single time point of 10s. I would like to see the HDX experiments being repeated at least 3 times at several time points (say 10s, 30s, 60s). Furthermore, include the deuterium uptake plots (and indicate standard deviations by error bars) for the representative peptides, which exhibit differential H/D exchange rates between longer and shorter protein constructs. At the same time, I have to say that HDX experiments are not critical for these studies because the authors chose the larger protein construct for x-ray crystallography; but if they want to include HDX results in the manuscript, these experiments need to be properly executed.

As a general comment, we thank the reviewers for their time and for their enthusiasm for the work.

Reviewer #1 (Remarks to the Author):

Qi and colleagues present a detailed study describing the crystal structure of the ELL1/2 and AFF4 binding surface using AFF4/ELL2 fusion proteins. This is followed by systematic point mutagenesis of AFF4 and ELL2 and in vitro and in vivo binding and functional studies using the HIV LTR as a model system. This is overall an interesting study that follows previous structural work of AFF4 and other SEC components.

Thank you for the positive comments.

However, despite considerable and detailed work on the mutants, the overall functional relevance of the described interaction interface seems low as point mutants only show interaction and functional defects when a second not very well described binding site in AFF4 is also deleted. In fact, in Figure 6a, deletion of this new, not well-described, second site nearly completely abrogates transactivation of the HIV promoter by the SEC while deletion of the here-characterized site has only a minor effect.

A full characterization of the C-terminal site is beyond the scope of the study. In brief, on the basis of the report of Yokoyama *et al.* and Cleary, *Cancer Cell* (2010), we believe the C-terminus is involved in heterodimerization between AFF1 and AFF4. The C-terminal domain bridges the mutant copy of AFF4 to wild-type endogenous AFF1 and the AFF1 ELLBow then binds to ELL2 in this model. This will be the topic of a future study from our groups. In the interim, we note that mutation of the ELL2 side of the binding site, shown in Fig. 7b, reduces function by ~4-fold.

In addition, the study is framed to be significant for HIV latency reversal and activation of the SEC but this is not supported by the current data.

We have added data using Jurkat 2D10 cells, a model system for HIV-1 latency, confirming that the mechanism we describe functions also in this model system. See the new Fig. 7c.

All data (using mutants) are geared towards disruption of binding, and it is unclear how activation can be achieved. However, this is prominently featured in the title, abstract and throughout the text and should be toned down. In the abstract, the authors highlight a binding pocket for a small molecule in the crystal structure but the only figure showing the relevance of this pocket is Supplemental figure S4, making it a minor feature of the manuscript in its current form.

Figure S4 has been moved in to the main manuscript and is now Figure 4.

Specific Comments

Figure 1: There is a "d" over the crystal structure in figure 1C. Figure 1E is not discussed in the text of the results section.

The error was actually in the legend. The **d** should have appeared one sentence earlier. It has been corrected. Fig. 1e is now cited in the discussion. Thank you for noticing these minor issues.

Figure 4: Please explain why the ELL2-Occ peptide was sumolyated. 4A: Y-axis needs to be labeled.

The ELL2-Occ construct is only moderately soluble and the sumo modification allows it to be concentrated to high levels in order to obtain the high concentration data points in the binding curves. Corrections and clarification have been applied to the figure and the figure legend. Thank you.

4A: Y-axis needs to be labeled.

Corrected. Thank you.

Please comment on biological replicates in the Figure legend.

The figure legend (now Fig. 5) states that the binding experiments, including new experiments added in Fig. 5d, were carried out in triplicate.

Figure 5:

-Here it is unclear why the 970-1163 mutant is introduced in order to assess the binding, please comment more on how this domain was discovered and how binding of the two interfaces is envisioned.

We now believe that the C-terminal domain is involved in heterodimerization with AFF1, and that it probably does not contain a second ELL1/2 binding site. Yokoyama *et al.* and Cleary, *Cancer Cell* (2010), showed that the C-terminal domain mediates heterodimerization between AFF1 and AFF4. Our results are consistent with Yokoyama *et al.*

-Why do E322H and P329G mutants exhibit decreased binding to ELL1 relative to ELL2? These mutants do not appear to have an effect in the FP experiments. What is special about ELL1 that confers differential binding? Please comment.

The ELL1 and ELL2 interfaces are relatively well conserved. Nevertheless, they are two different proteins, and even a single hydrogen bond or small number of van der Waals contacts could generate a difference in affinity big enough to manifest in the pull-down. The different binding behavior of ELL1 suggests that ELL1's EllBow-binding site could be subtly different from that of ELL2. For ELL1, perhaps E322 and P329 in the EllBow have more intimate contacts with ELL1 than ELL2 in the binding interface, or that the introduced H at 322 creates new unfavorable interactions unique to ELL1. Future structural studies of the ELL1-AFF4 interaction will be able to test this hypothesis.

-5B: The authors should use a lighter exposure for the ELL2-HA blots. There appears to be less ELL2-H608E/H559E-HA mutant. Why is the binding of CDK9 lost? Is it through loss of AFF4 binding? Please comment.

We have now provided a shorter exposure of the ELL2-HA panel (below the old ELL2-HA panel) in revised Fig. 6b. Compared to ELL2wt, ELL2-H608E/H559E-HA was expressed to a similar level and displayed somewhat decreased AFF-binding capacity. In contrast, even though ELL2-H608E/H559E/N619A/K625T was expressed at a higher level than wt and the double point mutant, it was almost completely defective in binding. Thus, the slight variations in sample loading in Fig. 6b do not affect the overall conclusion about the abilities of the two ELL mutants to bind to AFF1/4.

Why is the binding of CDK9 lost? Is it through loss of AFF4 binding? Please comment.

The reason for the loss of CDK9 binding is due to the fact that the association of P-TEFb with ELL2 is completely mediated by AFF4/1 within a SEC (Li *et al. MCB*, 2016). Thus, point mutations that disrupt the interaction of AFF4/1 with ELL2 are expected to also abolish the association with P-TEFb.

Please also comment on biological replicates.

Triplicate biological replicates in the luciferase assays means that three identical transfections were done in parallel in 3 wells of cells for each group. The three luciferase readings from the triple repeats were then averaged and had their standard deviations calculated for making the error bars. For western blot, an equal aliquot from each of the three whole cell lysates were pooled and loaded in each lane.

Figure 6:

Figure 6a shows basically that the 970-1163 mutation is more important for HIV-1 transcription than the 318-337 mutation or any of the point mutations herein (Supplemental Figure). The testing of the point mutants in the context of the 970-1163 mutation does not contribute anything because the 970-1163 mutation alone already decreases activation to almost baseline levels. It is unclear how these data support the main message of the paper.

It has been reported previously by Yokoyama et al. (Cancer Cell, 2010; 17:198) that the C-termini of AFF1 and 4 mediate the heterodimerization of the two proteins. The deletion of this region in the transfected FLAG-AFF4 could prevent it from dimerizing with endogenous AFF1, which could conceivably bring ELL2/1 into the immunoprecipitated complex independent of the ELLBow mutations present in the transfected FLAG-AFF4. Thus, the deletion of the AFF4 C-terminus is necessary for us to show the true effects of ELLBow point mutants on ELL2/1-binding *in vivo*. It would be interesting to further characterize the effects of the 970-1163 deletion on SEC formation, but it is beyond the scope of the current study.

Since the referee is not convinced that Supplementary Figure 5 contributes to the message of the paper, we removed it.

Figure 6b has no figure legend and the part of the figure labeled "dELL2" is not described in the text.

We apologize for the mislabeling. The Fig. 6 legend is for both panels A and B, because the same method and procedure were used in both panels. The confusion has been eliminated in the revised manuscript. dELL2 refers to a mutant HeLa cell line in which the ELL2 gene has been knocked out using the CRISPR/Cas9 system and this info has been added to the revised legend and methods section. The confirmation of ELL2 KO in dELL2 cells is shown in the new Figure S3.

As this is the only data that connect the structural findings with function via ELL2 it is critical to show the same experiments with AFF4 delta318-337 and AFF4 delta970-1163 and both to show that the ELL2 mutations decrease HIV transcription via interaction with AFF4318-337.

We hope that our discussion of heterodimerization of AFF4 with AFF1 via the C-terminal domain and the data of Yokoyama et al. 2010 will address this concern. The effects of AFF4 ELLBow mutants and ELL2-occ mutants have already been analyzed separately in Fig. 6, panels a and b. As discussed above, transfecting AFF4 Δ 318-337 together with ELL2 mutants in the luciferase assay may lead to AFF4's dimerization with the endogenous AFF1, resulting in the recruitment of endogenous AFF1-ELL2 to the LTR to complicate the analysis.

Reviewer #2 (Remarks to the Author):

The manuscript by Hurley and colleagues reports a straightforward structure-function analysis of the interaction between the transcription elongation factor ELL 1/2 and the scaffolding protein AFF 1/4. These proteins are components of the super elongation complex that transactivates HIV transcription upon recruitment by the tat-TAR complex in conjunction with P-TEFb. AFF1 (or 4) are intrinsically disordered proteins that interact with ELL 1/2 in the SEC. The N-terminal part of AFF4 has been determined in complex with Tat and most of P-TEFb. This manuscript adds to the description of the SEC by adding the interface of AFF4 with ELL2 to the already known interactions with P-TEFb and ENL. This was achieved by cleverly tethering the two proteins together to avoid aggregation of AFF 4. The most surprising observation is the extent to which the complex buries protein surface area. It further provides biochemical and functional (through transactivation assays) validation of the structural work and of how ELL is recruited to the SEC by traditional structure-guided mutagenesis. The experimental work is well done and presented.

Thank you for the many positive comments.

The only thing I found unclear in this otherwise well-written and well-executed study is the mention in the abstract of a binding site for a small molecule at the interface, followed up briefly on page 8 and a supplementary figure. This seems a relevant information both for understanding modulation of the SEC by metabolites and potentially for pharmaceutical targeting, but it is only very briefly brushed upon in the main text, and could be expanded. Otherwise, it might easily be missed by readers.

As described above, Fig. S4 has been moved to the main text as Fig. 4.

Reviewer #3 (Remarks to the Author):

This is a very nice manuscript, which reports crystal structure of the ELL2 C-terminal domain bound to its cognate AFF4 fragment. Furthermore, the authors demonstrate the functional significance of the ELL2-AFF4 interacting interface for HIV-1 Tat-mediated proviral transcription.

Thank you for the positive comments.

However, HDX studies are rather incomplete. The authors carried out one set of experiments using a single time point of 10s. I would like to see the HDX experiments being repeated at least 3 times at several time points (say 10s, 30s, 60s). Furthermore, include the deuterium uptake plots (and indicate standard deviations by error bars) for the representative peptides, which exhibit differential H/D exchange rates between longer and shorter protein constructs. At the same time, I have to say that HDX experiments are not critical for these studies because the authors chose the larger protein construct for x-ray crystallography; but if they want to include HDX results in the manuscript, these experiments need to be properly executed.

We are not sure there is any scientific value in carrying out the in depth HDX study. Since the reviewer considers these data not critical, we have removed them.

REVIEWERS' COMMENTS:

Reviewer #1 (Remarks to the Author):

In their rebuttal Qi and colleagues make a thorough attempt to address all criticisms. The additional 2D10 data and response to the additional information on the C-terminal AFF4 oligomerization domain are highly appreciated. However, the 2D10 data are not described in the text and should be described and referred to as Fig 7C in the main text. As no additional experiments with mutant AFF4(318-337) were performed that would have clarified whether the ELL2 mutations and mutant AFF4(318-337) work in the same pathway, the functional limitations of the study (little effect of mutant AFF4(318-337) possibly through C-terminal AFF4-AFF1 interaction while strong effect with C-terminal mutant alone) need to be more clearly described and implications also discussed.

In their rebuttal Qi and colleagues make a thorough attempt to address all criticisms. The additional 2D10 data and response to the additional information on the C-terminal AFF4 oligomerization domain are highly appreciated. However, the 2D10 data are not described in the text and should be described and referred to as Fig 7C in the main text.

Fig. 7c is now called out on pg. 9. Thank you for catching this omission.

As no additional experiments with mutant AFF4(318-337) were performed that would have clarified whether the ELL2 mutations and mutant AFF4(318-337) work in the same pathway, the functional limitations of the study (little effect of mutant AFF4(318-337) possibly through C-terminal AFF4-AFF1 interaction while strong effect with C-terminal mutant alone) need to be more clearly described and implications also discussed.

Please see the new sentence on page 10, second from end of the first paragraph of the discussion.